# Migration of Trivalent Praseodymium from Tombarthite Sewage by Microtubule Ultrafiltration Reactor with Organophosphorus in Fuel Oil

**DOI:** 10.3390/ijerph19159364

**Published:** 2022-07-30

**Authors:** Liang Pei, Liming Wang

**Affiliations:** 1Xinjiang Institute of Ecology and Geography, Chinese Academy of Sciences, Urumqi 830011, China; 2Institute of Geographic Sciences and Natural Resources Research, Chinese Academy of Sciences, Beijing 100101, China; 3University of Chinese Academy of Sciences, Beijing 100049, China; 4School of Environment and Chemistry Engineering, Xi’an Polytechnic University, Xi’an 710048, China; wangliming@xpu.edu.cn

**Keywords:** microtubule ultrafiltration reactor, organophosphorus, praseodymium(III), sewage section, enrichment section, migration percentage

## Abstract

A microtubule ultrafiltration reactor (MUFR), with an organophosphorus system containing a sewage section with buffer liquid acetic acid and an enrichment section with aqua fortis liquid and organophosphorus dissolved in fuel oil, has been studied for praseodymium(III) migration. Many factors of praseodymium(III) migration using MUFR need to be explored, including hydrogen ion molarity (or pH), cinit of praseodymium(III), the different ionic strengths of rare-earth mine sewage, the volume ratio of organophosphorus fuel oil and aqua fortis liquid (O/A), aqua fortis’ molarity, organophosphorus’ molarity, and the effects of different acid liquids in the enrichment section on praseodymium(III) migration with MUFR. The virtues of MUFR compared to conventional migration were explored. The effects of the hydrodynamic properties (stability and flow velocity) and UF system parameters (internal diameter of the microtubule, tubule shell thickness, void ratio), etc., on the mass migration performance of the MUFR process for praseodymium(III) migration were also studied. The experimental results show that the best migration prerequisites of praseodymium(III) were obtained as follows: an aqua fortis molarity of 4.00 mol/L, an organophosphorus molarity of 0.200 mol/L, an O/A of 0.6 in the enrichment section, and a pH value of 4.80 in the sewage section. The ionic strength of rare-earth mine sewage had no obvious effect on praseodymium(III) migration. When the cinit of praseodymium(III) molarity was 1.58 × 10^−4^ mol/L, the migration percentage of praseodymium(III) reached 95.2% in 160 min.

## 1. Introduction

Tombarthite metallic ions have been widely used in various industries. They can be used alone in some fields, or with other substances. Adding a certain amount of tombarthite metallic ions or tombarthite compounds can improve the properties of the alloy. Therefore, tombarthite elements are known as vitamins in the metallurgical industry [1,2]. For example, adding some tombarthite elements to steel can improve its plasticity, toughness, heat resistance, oxidation resistance and corrosion resistance. As another example, tombarthite metallic ions can be used to make fire alloys, permanent magnet materials, superconducting materials, dyeing materials, luminescent materials, trace element fertilizers, etc. [1,2,3]. In addition to being widely used in metallurgy, petrochemicals, glass, ceramics, fluorescent materials, electronic materials, medicine and agriculture, tombarthite metallic ions have gradually penetrated into many fields of modern science and technology. With the wide application of tombarthite elements in production and life, the migration and enrichment of tombarthite elements has become very necessary. In recent years, many institutions at home and abroad have studied this aspect [4,5,6,7,8]. There are many metallic ores in North China, many of which are rich in tombarthite metallic ions. At present, because of some political reasons, we are mining them one after another. However, the ores also produce some sewage that is difficult to treat. There is a praseodymium ore in the north provinces of China. There is much praseodymium sewage, which has some impact on the human and natural environment. The discharge of praseodymium sewage does not meet the required standards. We detected a certain amount of praseodymium in the plant roots around ore deposits. If this problem is not solved, humans eating these plants will cause damage to human viscera. The migration effect of traditional water treatment methods is not good. This study attempts to explore the method of extracting praseodymium, which provides theoretical support for the treating of praseodymium sewage in the future [9,10,11].

Film reactor technology is a new migration method that combines the characteristics of solvent migration and solid film migration. Compared with the conventional methods of solvent migration and liquid film technology, it has several potential virtues. Tombarthite metallic ion film reactor migration has the characteristics of being short process, with high speed, a high enrichment rate, low reagent consumption and low cost. In China, this research began in the early 1980s. The organic solvent for extracting tombarthite metallic ions in an ion exchange film reactor system is kerosene or ordinary fuel oil, the carrier can be various organic phosphonates, and the migration liquid can be various acids or bases. The tombarthite leaching liquid can be grouped, purified and separated according to the needs of the user [12].

Some scholars used stearic acid and phosphate as carriers to study the purification of tombarthite element Sm(III) in a conventional film reactor system, and established a migration model [13,14,15,16]. Others have also established a film reaction system of the tombarthite element Europium with polypropylene porous film as the supporting film and organophosphonic as the carrier, and established a mathematical model of the film reaction system and heavy metallic ions migration process [17]. Some people also use composite film reactors to treat the lanthanum in tombarthite sewage [18]. In recent years, some people [19,20] in China have studied the flat sandwich film reaction system, tested the permeability coefficient of extracting tombarthite metallic ions, compared the differences between film bodies with different materials and thickness in the migration process, and explored the migration rate and constancy of the film reaction system. In China, tombarthite resources are abundant, coming in many kinds and large quantities. It is a very arduous task to separate, purify and recycle them.

The filtration reactor system is simple to operate without adding expensive surfactants. However, due to the viscosity of organic liquids (organic solvents, migration agents and modifiers) in acid-base liquids, the contradiction between the decline of system constancy and high permeation flux has not been well solved [21]. In recent years, some experts at home and abroad have turned to explore new reactor structures. They hope to maintain the characteristics of film migration and overcome the inconstancy of ultrafiltration reactors. Therefore, a combined technology of a PAN ultrafiltration reactor and organophosphonic liquid was proposed [13,14]. It combines an ultrafiltration body or various chemical processes with an organophosphonic liquid, which can effectively overcome the problem of carrier leakage from the ultrafiltration body. In our previous work, we studied the migration of tombarthite in a dispersion-supported liquid film reactor, and the migration results are clear [22,23,24]. Based on the previous work of dispersion-supported liquid film reactor and hollow fiber strip dispersion reactor technology [25], this paper proposes a PAN microtubule ultrafiltration reactor with an organophosphorus system (MUFR), with aqua fortis as the migration agent and organophosphonic dissolved in kerosene as the mobile carrier. In the MUFR process, due to the wetting affinity between the hydrophobic microtubule and organic liquid, a very thin organic film is formed on the inner surface of the microtubule for separating the feed section and migration section [26]. The solute is selectively transported from the feed section to the migration section through microtubules. MUFR not only has the virtues of non-equilibrium mass transfer and an uphill effect, but it also avoids the drawbacks of conventional migration and liquid films, such as organic solute loss, the difficult operation of emulsification, de-emulsification steps, and so on.

At present, the application of MUFR in the migration of praseodymium(III) has not been reported. This paper has mainly explored and studied the feasibility of extracting praseodymium(III) using MUFR. The migration of tombarthite metallic ions was realized through membrane module design, carrier optimization and migration percentage control. The migration process of tombarthite metallic ions was studied, and a new method for extracting tombarthite metallic ions using MUFR was established. To sum up, there are still some gaps in the ore sewage treatment technology, which we will fill in the field of ore sewage treatment using praseodymium.

## 2. Materials and Methods

### 2.1. MUFR Migration Process

The enrichment section and feeding section are both 500 mL measuring cups. There is a pump for the entry and exit of liquid via the ultrafiltration membrane. The other parameters of the pump and ultrafiltration membrane can be fine-tuned. The sewage section is prepared by dissolving a certain amount of Pr(CH_3_COO)_3_·4H_2_O in acetate buffer liquid with an adjusted acidity value. The enrichment section takes organic phosphonate as the carrier and is dissolved in fuel oil. The extract is 3.0 mol/L^−1^ aqua fortis liquid. We soak the microtubule ultrafiltration body with organic liquid for at least 48 h to completely fill the pores of the microtubule with organophosphorus liquid. In these experiments, there are two operation modes: (I) The extract enters the system through the outer wall of the ultrafiltration tube, and the sewage enters the system through the ultrafiltration tube. (II) The extract enters the system through the ultrafiltration tube, and the sewage enters the system through the outer wall of the ultrafiltration tube. Both fluids are transported in opposite directions and in a single path process. Generally speaking, due to the weak surface activity of the migration liquid, the use of a stirrer can help to mix the migration liquid with the organophosphonic liquid and form droplets of the migration liquid in a continuous organic liquid. This is referred to as the enriched section. During the experiment, there was a constant supply of organophosphonic liquid, that is, the enriched organophosphonic liquid was in contact with the micropores of UF. The continuous supply of this organophosphonic liquid ensures the stable and continuous operation of UF. In this way, the complex of tombarthite metallic ions and carrier can diffuse into the interface between the migration liquid and the organic liquid in UF pores. Therefore, the direct contact between the migration liquid and the organophosphonic liquid provides effective mass transfer for tombarthite metallic ions migration. Secondly, once the migration of the target species is completed, the agitator used for migration will not work. The organic liquid suspension is allowed to separate the enrichment section into two sections: an organophosphonic liquid that can easily wet the carrier hole, and a migration liquid containing concentrated tombarthite metallic ions. The concentrated extract is the product of this process. Figure 1 shows the experimental device of the MUFR process.

In order to prevent the leakage of the organophosphonic liquid from the film hole, the pressure on both sides can be controlled through the flow velocity to make the pressure outside the ultrafiltration tube slightly positive and form a stable interface.

Based on our previous experiments, a stable praseodymium(III) molarity distribution was obtained in the sewage section and enrichment section after 22 min, and then a stable mass transfer performance was realized. Similar results were found when the enriched section flowed through the tube side. In later experiments, the initial stabilization time was set to more than 30 min to obtain more reliable results.

### 2.2. MUFR Migration Process Principle

Figure 2 shows the principle of the MUFR process, which describes the molarity change and migration process. The migration process involves various equilibrium reactions, which are described as follows [27]:

(a) Praseodymium(III) diffuses from the sewage section to interface X.

(b) On the sewage-side interface of MUFR, praseodymium(III) is extracted from the sewage liquid with the carrier organophosphorus (which can be (HR)_2_) in fuel oil, which can be expressed as [27]:(1)$${\mathrm{Pr}}_{f}^{3+}+3{\left(\mathrm{HR}\right)}_{2,\mathrm{org}}\overset{K_{1}}{\underset{K_{-1}}{⇄}}\mathrm{Pr}R_{3}\cdot 3\mathrm{HR}+3H_{f}^{+}$$
where f and org represent sewage and organophosphonic liquid, respectively; (HR)_2_ shows that the organic phosphorus in the fuel oil mainly exists in the form of dimer; *K*_1_ and *K*_−1_ represent the forward and reverse reaction percentage constants at the interface between the sewage section and the ultrafiltration microtubule.

(c) The metallic ions complex (PrR_3_·3HR) was extracted with film X-Y.

(d) At the migration-side interface of MUFR, the PrR_3_·3HR and metallic ions praseodymium(III) dissolved in organophosphonic liquid are extracted by the migration agent.

The migration reaction on the other side of the microtubule is as follows:(2)$$\mathrm{Pr}R_{3}\cdot 3\mathrm{HR}+3H_{s}^{+}\overset{K_{2}}{\underset{K_{-2}}{⇄}}{\mathrm{Pr}}^{3+}+3{\left(\mathrm{HR}\right)}_{2,\mathrm{org}}$$
where s represents the enrichment section; *K*_2_ and *K*_−2_ represent the forward and reverse reaction percentage constants at the interface between the organophosphonic liquid and the enrichment section, respectively.

(e) The carrier organophosphorus returns from Y to X.

In this mechanism, the migration of praseodymium(III) via MUFR will be described by considering the diffusion coefficient of praseodymium(III). This is because the complex reaction between praseodymium(III) and organophosphate at the interface is much faster than the reaction with sewage and the microtubule [27].

In order to establish the model, the following assumptions are made:(1)Praseodymium(III) moves only in the form of the PrR_3_·3HR complex in organophosphorus liquid;(2)The convection in the ultrafiltration microtubule has no net flow;(3)Metallic ions only react with organophosphorus liquid at the microtubule interface;(4)The organophosphorus monomer and dimer are always in equilibrium in the whole enrichment section;(5)We found that the solubility of organophosphorus liquid in acidic liquid is negligible. Therefore, it is assumed that the molarity of organophosphorus in MUFR remains unchanged [19].

In this model, that can be considered that the migration of praseodymium(III) in MUFR includes four consecutive steps. If the migration process can be described by Fick’s law, the migration flux of each step is as described in [24,25,26,27,28,29].

The flow of sewage can be written as:(3)$$J_{f}=\frac{D_{f}}{d_{f}}(c_{f}-c_{\mathrm{fi}})$$
where *J*_f_, *D*_f_, *d*_f_, *C*_f_ and *C*_f__i_ respectively represent the migration flux in the sewage section, the diffusion coefficient of praseodymium(III) in the microtubule, the thickness of the interface between the sewage section and the microtubule, the molarity of praseodymium(III) in the sewage section, and the molarity of praseodymium(III) in the interface between the sewage section and the microtubule.

The migration equilibrium constant *K*_ex_ in Formula (1) can be expressed as:(4)$$K_{\mathrm{ex}}=\frac{{\left[H^{+}\right]}^{3}c_{\mathrm{mf}}{}^{0}}{c_{\mathrm{fi}}{\left[{\left(\mathrm{HR}\right)}_{2}\right]}^{3}}=\frac{K_{d}{\left[H^{+}\right]}^{3}}{{\left[{\left(\mathrm{HR}\right)}_{2}\right]}^{3}}=\frac{8K_{d}{\left[H^{+}\right]}^{3}}{{\left[\mathrm{HR}\right]}^{3}}$$
where m represents organophosphonic microtubule and *K*_d_ represents the distribution ratio of praseodymium(III).

The migration flux of the interface layer between the sewage section and the organophosphonic microtubule can be written as:(5)$$J_{\mathrm{mf}}=K_{1}c_{\mathrm{fi}}-K_{-1}c_{\mathrm{mf}}{}^{0}$$
where *J*_mf_ and *c*_mf_^0^ represent the migration flux and praseodymium(III) molarity in the sewage section and the organophosphonic microtubule interface layer.
(6)$$J_{m}=\frac{D_{m}{}^{0}}{d_{m}}(c_{\mathrm{mf}}{}^{0}-c_{\mathrm{ms}}{}^{0})$$
where *J*_m_, *c*_ms_^0^, *D*_m_^0^ and *d*_m_ respectively represent the migration flux at the reaction interface, the molarity of praseodymium(III) in the organophosphonic and microtubule interface layer, the migration coefficient of praseodymium(III) in the microtubules, and the thickness of the microtubules.

In view of the different characteristics of different microtubules, the curvature and porosity of microtubules have an impact on the migration flux of microtubules. The greater the porosity of the microtubules, the higher the flux, and the greater the curvature of microtubules, the lower the flux. Therefore, considering the correction coefficient, Equation (6) can be rewritten as:(7)$$J_{m}=\frac{D_{m}{}^{0}\varepsilon }{d_{m}\tau }(c_{\mathrm{mf}}{}^{0}-c_{\mathrm{ms}}{}^{0})$$
where *τ* and *ε* represent the curvature and porosity of microtubules, respectively.

In the internal steady-state reaction system under ideal conditions, all the above individual migration fluxes can take the same value [12,15,16,17,18,19].
*J*_f_ = *J*_m_ = *J*_mf_ = *J*_s_
(8)

where *J*_s_ represents the migration flux of the dispersed phase.

Combining Formulas (3), (4) and (6)–(8), the following formula can be obtained:(9)$$J=\frac{1}{\frac{d_{f}}{D_{f}}+\frac{8d_{m}\tau {\left[H^{+}\right]}^{3}}{D_{m}{}^{0}\varepsilon K_{\mathrm{ex}}{\left[\mathrm{HR}\right]}^{3}}}c_{f}$$

According to the definition of the permeability coefficient, the migration flux of the microtubule can also be written as:(10)$$J=P_{C}\cdot c_{f}=-\frac{V_{f}}{\varepsilon A}(\frac{dc_{f}}{dt})$$
where *V*_f_ represents the volume of sewage, *A* represents the effective area of the microtubule and *J* represents the flux of the microtubule.

*P*_C_ stands for permeability coefficient, which can be defined as:(11)$$P_{C}=\frac{1}{\frac{d_{f}}{D_{f}}+\frac{8d_{m}\tau {\left[H^{+}\right]}^{3}}{D_{m}{}^{0}\varepsilon K_{\mathrm{ex}}{\left[\mathrm{HR}\right]}^{3}}}$$

*d*_f_/*D*_f_ and *d*_m_/*D*_m_^0^ are defined as follows:(12)$$\delta_{f}=\frac{d_{f}}{D_{f}}$$
(13)$$\delta_{m}=\frac{d_{m}}{D_{m}{}^{0}}$$

Equation (11) can be simplified as:(14)$$\frac{1}{P_{C}}=\frac{8\delta_{m}\tau {\left[H^{+}\right]}^{3}}{\varepsilon K_{\mathrm{ex}}{\left[\mathrm{HR}\right]}^{3}}+\delta_{f}$$

In Equation (14), *ε*, *τ* and *K*_ex_ are constants. Through the migration experiment, *K*_ex_ 1.5 × 10^−10^ was obtained. Under the same organic phosphorus molarity, 1/*P*_C_ has a linear relationship with [H^+^]^3^. Therefore, the migration coefficient of praseodymium(III) in microtubules and the thickness of the sewage microtubule interface can be obtained by the linear slope method. Similarly, with the same acidity of the sewage section, the relationship between 1/*P*_C_ and [HR]^3^ was found to be linear.

Then, we can derive *δ*_f_ and *δ*_m_. The *D*_m_^0^ and *d*_f_ values can also be obtained according to Equation (14), in combination with Equations (12) and (13).

The migration percentage of praseodymium(III) was obtained by measuring *dc*_f_/*dt*.

We consider and integrate Equation (10) as follows:(15)$$\ln\frac{c_{t}}{c_{0}}=-\frac{\varepsilon A}{V_{f}}P_{C}\cdot t$$
(16)$$\eta =1-e^{-\frac{\varepsilon A}{V_{f}}P_{C}\cdot t}\;\mathrm{or}\;\ln(1-\eta )=-\frac{\varepsilon A}{V_{f}}P_{C}\cdot t$$
(17)$$\int_{c_{{}_{0}}}^{c_{t}}\frac{dc_{f}}{c_{f}}\cdot \frac{-V_{f}}{\varepsilon AP_{C}(t^{*})}=t-t_{0}$$
where *c*_t_ and *c*_0_ represent the molarity of praseodymium(III) in the sewage at *t* = 0 and *t* = *t*, respectively. *t** represents the average time. Equation (15) shows that the value of *P*_C_ is the slope of the straight line obtained under different operating prerequisites.

Combining Equations (10)–(17), the following equation can be obtained:(18)$$\eta =1-e^{\frac{8\delta_{m}\tau V_{f}{\left[H^{+}\right]}^{3}}{\varepsilon^{2}K_{\mathrm{ex}}At{\left[\mathrm{HR}\right]}^{3}}+\frac{\delta_{f}V_{f}}{\varepsilon At}}$$

According to Fick’s second law, Equation (18) can also be written as follows [13]:(19)$$J|{}_{t}=\frac{D_{m}^{0}K_{\mathrm{ex}}{\left[{\left(\mathrm{HR}\right)}_{2}\right]}^{3}c_{f}}{d_{m}{\left[H^{+}\right]}^{3}}\left(1-\frac{d_{m}{}^{2}}{6D_{m}^{0}t}\right)$$
where *ε*, *τ* and *K*_ex_ are constants. *η* represents the migration percentage of praseodymium(III) through MUFR. *J* represents the migration flux of praseodymium(III). *c*_f_ is the molarity of the sewage section. *t* is the time. *D*_f_ represents the movement coefficient of praseodymium(III) in microtubules. *d*_f_ represents the thickness of the interface layer between the waste water section and the film phase. *D*_m_^0^ represents the coefficient of praseodymium(III) in the microtubules. *d*_m_ represents the thickness of microtubules.

### 2.3. Reagent

The Pr(CH_3_COO)_3_·4H_2_O, Arsenazo III, HNO_3_, NaH_2_PO_4_, Na_2_HPO_4_, CH_3_COONa and CH_3_COOH used in this study are of analytical grade. 2-ethylhexyl phosphate Mono-2-ethylhexyl ester (organophosphorus) is a commercial migration agent (purity > 96%) that was used without further purification. The fuel oil is washed with concentrated sulfuric acid and distilled at 180–215 °C.

### 2.4. Preparation of Liquid

Praseodymium(III) stock liquid: Dissolve Pr(CH_3_COO)_3_·4H_2_O in 0.8 mol/L aqua fortis to prepare praseodymium(III) stock liquid, and analyze this with Arsenazo III as the chromogenic agent.

Praseodymium(III) sewage liquid: After adding the calculated amount of CH_3_COONa and CH_3_COOH or Na_2_HPO_4_ and NaH_2_PO_4_, dilute a certain amount of praseodymium(III) stock liquid to a given degree with 0.04 mol/L aqua fortis liquid.

Arsenazo III stock liquid: Arsenazo III stock liquid is prepared by dissolving Arsenazo III powder in deionized water.

Back migration liquid: dissolved with deionized water and diluted to the required amount of aqua fortis with known molarity.

### 2.5. Experimental Materials and Determination

All the experiments are self-designed systems. The microtubule ultrafilter is a laboratory-scale version with two 0~200 mL·min^−1^ pumps and flowmeters. This makes it possible to evaluate the performance without preparing a large amount of sewage, back migration agent and organophosphonic liquid, and avoids the influence of non-ideal flow on both sides of the microtubule. The microtubule ultrafiltration reactor used a commercial PAN module, with nominal porosity of 48%, bending degree of 2.15, effective module length of 28 cm, module internal diameter of 3.3 cm and number of microtubules of 19. The internal diameter, outer diameter, thickness and effective film area of the microtubule were 2.85 mm, 3.18 mm, 0.27 mm and 181 cm^2^, respectively.

The ion molarity of the sample containing only praseodymium(III) in the feed liquid was analyzed with a UV-1200 spectrophotometer and Arsenazo III as the developer (detection wavelength: 653 nm).

## 3. Results and Discussion

### 3.1. Constancy of MUFR

In order to determine the constancy of MUFR compared with conventional supported liquid film, the variation trends of praseodymium(III)’s molarity in the sewage section and enrichment section with time under long-term fixed operation prerequisites were studied.

In this experiment, two operation modes have been studied. In mode 1, the enrichment section enters through the outside of the microtubule wall and the sewage section enters through the inside of the microtubule. The flow velocities of the sewage section and enrichment section were 0.017 m/s and 0.008 m/s, respectively. The flow velocity of the two sections was maintained in the experiment.

The selected hypothetical experimental prerequisite was to adjust the pH value to 4.60 under the specific pH value of the sewage section. In the sewage section, the cinit of praseodymium(III) was 1.84 × 10^−4^ mol/L, the volume ratio (O/A) of organophosphonic liquid/migration liquid was 0.3, the molarity of aqua fortis in the enriched section was 3.10 mol/L, and the molarity of organophosphorus was 0.210 mol/L. The results are shown in Figure 3 and Figure 4. After 30 min, the change trend of praseodymium(III) molarity was stable, so we took samples two at a time, once every 30 min and once every 60 min. The praseodymium(III) molarity and constancy of the enrichment section gradually decreased when a conventional supported liquid film was used, and the praseodymium(III) molarity of the sewage section and the enrichment section remained stable when MUFR was used. This is because the organophosphonic in the conventional supporting liquid film was gradually lost, while the MUFR with organophosphonic liquid can continuously supplement the system with organophosphonic liquid. Therefore, we can conclude that the constancy of MUFR is better than that of the conventional supported liquid film.

### 3.2. Effect of Flow Velocity of Sewage Section and Enrichment Section

In order to study the migration mechanism in MUFR, and determine the main section of the total migration resistance, it is necessary to study the hydrodynamic characteristics of the system. The flow velocity of sewage and enrichment plays an important role in the migration of metallic ions from the sewage to the back migration liquid. In the conventional supported liquid film, the carrier will be gradually washed away by the high-speed flowing liquid. In MUFR, the organophosphonic liquid helps to provide a carrier for microtubules under high-speed flow prerequisites. Therefore, this section studies the effect of the flow velocity of the sewage section and the enrichment section on the migration rate of praseodymium(III). All other parameters, such as pH value, the cinit of praseodymium(III) in sewage, the volume ratio of organophosphonic liquid/acid liquid (O/A), and the organophosphorus molarity, were adjusted to 4.60 and 1.76 × 10^−4^ mol/L, 0.50 and 0.220 mol/L respectively. The effects of the flow velocity of the sewage section and the enrichment section on the migration percentage of praseodymium(III) are shown in Figure 5 and Figure 6.

We find that in the conventional supported liquid film, the high-speed flowing liquid will gradually wash away the organophosphonic carrier, so the higher the flow velocity, the lower the migration rate. In MUFR, under the prerequisite of a high-speed flow, organophosphonic liquid helps to provide a carrier for microtubules. Therefore, the outer wall liquid flow velocity of the microtubule has no significant effect on the migration rate of praseodymium(III). The flow velocity of sewage liquid in the microtubule has little effect on the migration rate of praseodymium(III). This is because the diffusion rate of the sewage liquid in the inner boundary layer of the microtubule is an important rate control step in the whole migration process [28,29]. The flow velocity of the organic phosphine liquid is about 9.5 mL/min in the outer wall of the micro tube, and the flow velocity is limited to about 1.5 mL/min.

### 3.3. Effect of Sewage pH

According to the mechanism of the migration process, the molarity difference between the sewage section and the enrichment section is the driving force of the mass migration process [14]. Therefore, in the sewage section, the lower the H^+^ molarity, the stronger the driving force of the mass migration process. More power (a strong hydrogen ion concentration difference) will promote the migration flux of praseodymium(III). Similarly, the higher the pH value of sewage, the higher the migration flux of praseodymium(III). The effect of sewage pH on praseodymium(III) migration was studied in the range of pH 2.40 to 5.00. In the sewage section, the cinit of praseodymium(III) is 1.58 × 10^−4^ mol/L. In the enrichment section, the molarity of aqua fortis liquid is 4.00 mol/L, the volume ratio of organophosphonic liquid/acid liquid (O/A) is 0.50, and the molarity of organophosphorus is 0.120 mol/L. The results are shown in Figure 7. When the pH value of sewage increased from 2.40 to 5.00, the migration percentage of praseodymium(III) increased, and the maximum migration percentage observed at pH 4.80 within 160 min was 89.5%. When the pH value of sewage is higher than 4.80, the migration percentage of praseodymium(III) is unstable within 160 min due to sewage emulsification. All previous studies [20,24] proposed the influence of pH value on the section coefficient of the migration process. This is because of the regeneration effect of the organophosphonic liquid on the microtubules. When the diffusion mobility of praseodymium(III) ions is definitive under specific experimental conditions, the migration process is mainly controlled by the mass transfer driving force caused by the distribution balance [18,22,23]. We chose the pH value of 4.80 as the optimal feeding prerequisite for our study.

### 3.4. Effect of Acid Molarity in Enrichment Section

The stripping reaction of the enriched section plays an important role in extracting metallic ions from sewage at the stripping stage. Therefore, this section studies the effect of the molarity of enriched aqua fortis on the migration flux of praseodymium(III). All the other parameters, such as pH value, the cinit of praseodymium(III) in sewage, the volume ratio of organophosphonic liquid/acid liquid (O/A) and the organophosphorus molarity, were adjusted to 4.80, 1.58 × 10^−4^ mol/L, 0.50 and 0.120 mol/L, respectively. The effect of the aqua fortis molarity of the enriched section on praseodymium(III) migration flux is shown in Figure 8. The results show that the migration flux of praseodymium(III) increased with the increase in the molarity of enriched acid. It can be seen that the effective acid molarities for migration are 3.00 mol/L, 4.00 mol/L and 5.00 mol/L, which make the migration flux of praseodymium(III) about 4.99 × 10^−7^ mol/(s·m^2^), 6.07 × 10^−8^ mol/(s·m^2^) and 5.42 × 10^−8^ mol/(s·m^2^), respectively.

Increasing the molarity of aqua fortis liquid from 1.00 mol/L to 2.00 mol/L had no significant effect on the migration flux of praseodymium(III). This is because the number of praseodymium(III) complexes and the molarity of the organophosphonic liquid extracted by microtubules per unit time are definitive. Considering the control of acidity and the increase in migration flux, we chose 4.00 mol/L as the optimal acid molarity of the concentrated section for the later experiments.

### 3.5. Effect of Volume Ratio of Enriched Section (O/A)

The effect of the volume ratio of organophosphonic liquid to acid liquid (O/A) in the enrichment section on the migration of praseodymium(III) was studied. It is assumed that the selected experimental prerequisites include the specific pH value of the sewage section, which was adjusted to 4.80. The cinit of praseodymium(III) in the sewage was adjusted to 1.58 × 10^−4^ mol/L, the acid molarity of the enrichment section was adjusted to 4.00 mol/L, and the molarity of organophosphonic was adjusted to 0.120 mol/L. The effect of the volume ratio (O/A) of the enriched section on the separation of praseodymium(III) is shown in Figure 9. The volume ratio increased from 0 to 2.00. It can be seen that the most effective volume ratio was 0.60, which made the migration flux of praseodymium(III) much higher than that of other substances.

This shows that the migration flux of praseodymium(III) increases with the increase in the volume ratio of the enriched section. When the volume ratio of the enriched section increases, the organophosphonic liquid is obviously dispersed in the microtubules. Therefore, the probability of contact between organophosphorus liquid and praseodymium(III) increases. In this way, the mixing of microtubules and enriched sections provides an additional regeneration percentage to the migration surface and the microtubule surface carrier. Thus, the efficiency of extracting the targeted heavy metallic ion complex from organophosphonic liquid to acidic liquid is greatly improved. When the volume ratio increases to a certain extent, the flux decreases due to the reduction in H+ in the enriched section [17]. In the following experiment, we chose 0.60 as the optimal volume ratio (O/A) of the organophosphonic liquid and acid liquid.

### 3.6. Effects of Different Acid Liquids as Stripping Agents

The concentrated acid liquid plays an important role in recovering metallic ions from sewage in the stripping stage. Therefore, the effects of different acids on the migration of praseodymium(III) were studied. We adjusted the pH value of the sewage section, the cinit of praseodymium(III), the volume ratio (O/A) of the enrichment section and the molarity of organic phosphorus to 4.80, 1.58 × 10^−4^ mol/L, 0.60 and 0.100 mol/L, respectively. The effects of using different acids in the enrichment section on the migration rate of praseodymium(III) are shown in Figure 10. Under the same acidity prerequisites, hydrochloric acid, sulfuric acid and aqua fortis were used as stripping agents, respectively. It was found that aqua fortis was the most effective stripping agent in this study.

### 3.7. Effect of Praseodymium(III) Cinit in Waste Water Section

When the cinit of praseodymium(III) was in the range of 8.00 × 10^−5^ mol/L to 2.00 × 10^−4^ mol/L, the effect of cinit on the migration rate of praseodymium(III) was studied. The pH value of the sewage section was adjusted to 4.80, the volume ratio (O/A) was adjusted to 0.60, the molarity of aqua fortis in the enrichment section was adjusted to 4.00 mol/L, and the molarity of organophosphonic liquid was 0.20 mol/L. The results are shown in Table 1. With the praseodymium(III) cinit in the sewage section increased from 1.29 × 10^−4^ mol/L to 2.00 × 10^−4^ mol/L, the migration rate of praseodymium(III) decreased. This is because, when the interface between the sewage section and the microtubule is determined, the amount of organophosphorus passing through the microtubule is certain. That is, in this migration process, the amount of praseodymium(III) extracted is determined [17]. When the praseodymium(III) cinit was 8.00 × 10^−5^ mol/L, 1.02 × 10^−5^ mol/L, 1.29 × 10^−5^ mol/L, 1.58 × 10^−4^ mol/L or 2.00 × 10^−4^ mol/L, the migration rate in 100 min was 91.41%, 73.2%, 72.3%, 61.2% or 49.3%, respectively.

### 3.8. Effect of Organophosphonic Molarity

The molarity of organophosphonic liquid in the microtubules and enriched section also has an important effect on the migration of praseodymium(III). The effect of organophosphorus acid molarity on the migration rate of praseodymium(III) was studied in the range of organophosphorus acid molarity from 0.06 mol/L to 0.220 mol/L. I we adjust the pH value to 4.80, the cinit of praseodymium(III) in the sewage is 1.58 × 10^−4^ mol/L, the volume ratio (O/A) of the concentrated section is adjusted to 0.60, and the aqua fortis molarity is also adjusted to 4.00 mol/L. The results are shown in Figure 11. The migration rate of praseodymium(III) increased with the increase in film carrier molarity from 0.06 mol/L to 0.220 mol/L. However, when the molarity of organophosphonic increased from 0.200 mol/L to 0.220 mol/L, the migration rate of praseodymium(III) did not increase significantly. In the range of organophosphonic molarity from 0.060 mol/L to 0.200 mol/L, the effectiveness of organophosphonic liquid on the microtubule organic interface of the sewage section increased with the increase in organophosphonic molarity. Therefore, the chemical reaction is balanced to the left. Similarly, when the molarity of organophosphonic liquid becomes low, the equilibrium moves to the right [19]. When the molarity of the organophosphonic liquid increases significantly, the migration flux of praseodymium(III) will no longer increase with time [17]. When the molarity of the organophosphonic was 0.100 mol/L, 0.130 mol/L, 0.200 mol/L or 0.220 mol/L, the migration rates were 86.1%, 93.1%, 95.2% and 93.5%, respectively. The molarity of organophosphonic liquid is directly proportional to the molarity of praseodymium(III) in the microtubules. When the molarity of the organophosphonic liquid in the microtubule is higher than that of the praseodymium(III) in sewage, no praseodymium(III) reacts with the excess organophosphonic liquid, so the increase in the praseodymium(III) migration rate will slow down. This shows that when the initial molarity of praseodymium(III), the action area, and the time of the film are fixed, the amount of organophosphorus extracted by the film per unit of time is determined. The optimum molarity of organophosphonic liquid is 0.200 mol/L. Under these conditions, the migration rate of praseodymium(III) reached 95.2% within 160 min.

### 3.9. Effect of Ionic Strength in Sewage

In this section, the effect of the ionic strength of sewage on the migration rate of praseodymium(III) is studied. The results are shown in Figure 12. The results show that the ionic strength had no significant effect on the migration flux of praseodymium(III). Because the aqua fortis effect when surrounded by organophosphonic liquid is weak, the ionic strength of the enriched section can be ignored. When the ion molarity of the sewage is low, the ionic strength of the two sections can be ignored. Compared with other migration technologies, the prerequisites of operation can be further simplified [19].

### 3.10. Reuse of Organophosphonic Liquid

Under the optimum conditions, the reuse of organophosphonic liquid was studied. It can be seen from Figure 13 that the organophosphonic liquid in MUFR can be reused many times before re-migration with strong acid after each experiment. In the four experiments, the organophosphonic liquid in the MUFR was reused many times, and the variation trend of the migration rate was stable. After five experiments, the migration rate of praseodymium(III) decreased gradually.

### 3.11. Reuse of PAN Microtubules

The reuse of PAN microtubules was studied under the optimum conditions. The organophosphonic liquid was reused only four times. In the fifth experiment, we used a new organophosphonic liquid or an organophosphonic liquid purified by strong acid stripping. The results are shown in Figure 14. Over nine experiments, the MUFR microtubules with an enrichment section were reused many times, and the variation trend of the migration rate was stable. However, when there was no organophosphonic liquid in the MUFR enrichment section, the variation trend of the migration rate was unstable, and the migration rate gradually decreased after three experiments. From this study, we can also conclude that the constancy of the MUFR system is better when the organophosphonic liquid is increased in the enrichment section.

### 3.12. Retention in PAN Microtubules and Stripping Effects

Under the optimum conditions, the retention in the PAN microtubules and the stripping effects were studied. We adjusted the pH value to 4.80, the cinit of the praseodymium(III) in the sewage to 1.58 × 10^−4^ mol/L, the volume ratio (O/A) to 0.60, and the molarity of aqua fortis liquid in the concentrated section to 4.00 mol/L. We adjust the molarity of the organophosphonic liquid to 0.200 mol/L. According to the molarity of praseodymium(III) in the sewage and stripping stages, the molarity of praseodymium(III) in PAN microtubule can be obtained, and then the stripping effect of the enrichment section and the retention phenomenon of the PAN microtubule can be obtained. The results are shown in Figure 15. We can conclude that the retention rate of praseodymium(III) in the PAN microtubules gradually decreases with the extension of time, because the stripping speed in the migration process is faster than the complexation speed.

### 3.13. Effect of Structural Parameters of PAN Ultrafiltration Microtubules

The effects of the microtubule’s structural parameters (tube internal diameter, tube thickness and porosity) on the mass migration performance of praseodymium(III) migration using MUFR were also studied. Six polyacrylonitrile (PAN) microtubules were selected, and additional information on these modules is listed in Table 2.

#### 3.13.1. Effect of Microtubule Thickness

The microtubule thickness will affect the thickness of organophosphonic interface layer in MUFR module and the migration rate of the praseodymium(III) complex. Microtubule modules M1, M2 and M3 were selected to study the effect of microtubule thickness on praseodymium(III) migration under the optimal migration prerequisites. The results are shown in Figure 16. We can conclude that the thicker the microtubule, the lower the migration rate of praseodymium(III). The main reason may be that the thicker the microtubule, the lower the migration rate of the praseodymium(III) complex.

#### 3.13.2. Effect of Microtubule Internal Diameter

The internal diameter of the microtubule will affect the flow state of the liquid and the migration rate of praseodymium(III) in an MUFR system. The effect of the microtubule’s internal diameter on the migration of praseodymium(III) was studied under the optimum migration conditions. The results are shown in Figure 17. The results show that the larger the internal diameter of the microtubules, the higher the migration rate of praseodymium(III). The main reason may be that the larger the internal diameter of the microtubule, the more organophosphonic liquid can enter the microtubule system. Therefore, the presence of more organophosphonic liquid can make the stripping and complexation reaction more effective.

#### 3.13.3. Effect of Microtubule Porosity

The porosity of the microtubules has an important effect on the migration percentage of praseodymium(III) complexes. Microtubule modules M1 and M6 were selected to study the effects of microtubule porosity on the migration of praseodymium(III) under the optimal migration prerequisites. The results are shown in Figure 18. We can conclude that the larger the porosity of the microtubules, the higher the migration rate of praseodymium(III). The main reason may be that the greater the porosity of the microtubules, the larger the effective migration area of the microtubules.

## 4. Conclusions

It has been found that MUFR systems using organophosphates on tombarthite metallic ions can selectively separate praseodymium(III) from acidic media. We can draw the following conclusions. The optimum migration prerequisites of praseodymium(III) in the MUFR system are as follows: the molarity of aqua fortis liquid is 4.00 mol/L, the volume ratio (O/A) of organophosphonic liquid to acid liquid is 0.60, the molarity of organophosphonic in the enrichment section is 0.200 mol/L, and the pH value of the sewage section is 4.80. When the initial molarity is 1.58 × 10^−4^ mol/L, the migration effect of praseodymium(III) is very obvious under the best prerequisites. With a migration time of 160 min, the migration rate of praseodymium(III) reached 95.2%.

The constancy of MUFR is better than that of conventional supported liquid film. The outer wall’s liquid velocity in the microtubule has little effect on the migration rate of praseodymium(III), and the sewage velocity in the microtubule has little effect on the migration rate of praseodymium(III). The organophosphonic liquid in MUFR can be reused more than four times after re-migration with strong acid after each experiment. Across nine experiments, the MUFR module with an enrichment section can be reused many times, and the variation trend of the migration rate is stable. With the extension of time, the retention rate of praseodymium(III) in the microtubules decreased gradually. The migration rate of praseodymium(III) increases with the decreasing of film thickness, the increase in microtubule internal diameter, and the increasing of microtubule porosity, respectively. This study attempted to explore a system of praseodymium extraction. It provided theoretical support for the purification of praseodymium ore sewage in the future. The future research direction will be the industrial application of the technology system. We still need to make slightly larger devices for in-depth research, but the premise is to reduce the price of membrane materials. This necessitates our country giving sufficient support in making a breakthrough in membrane materials, along with the development of environmental protection technologies and support policies.

## Figures and Tables

**Figure 1 ijerph-19-09364-f001:**
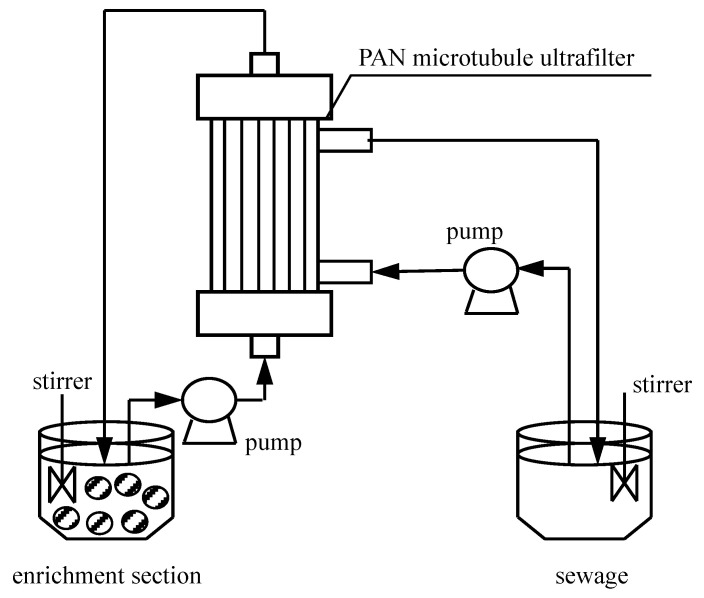
Installation of PAN microtubule ultrafiltration reactor process.

**Figure 2 ijerph-19-09364-f002:**
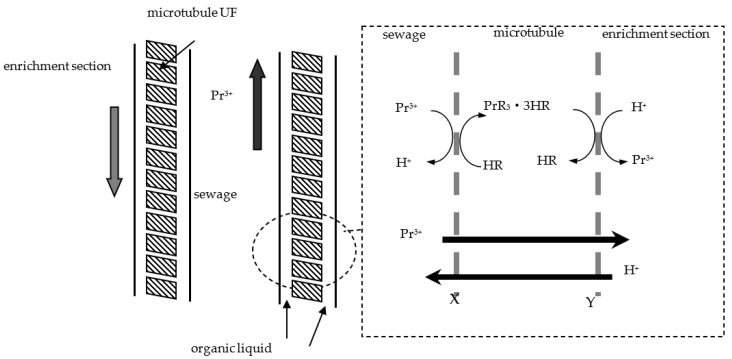
Schematic principle and description of Pr^3+^ migration MUFR.

**Figure 3 ijerph-19-09364-f003:**
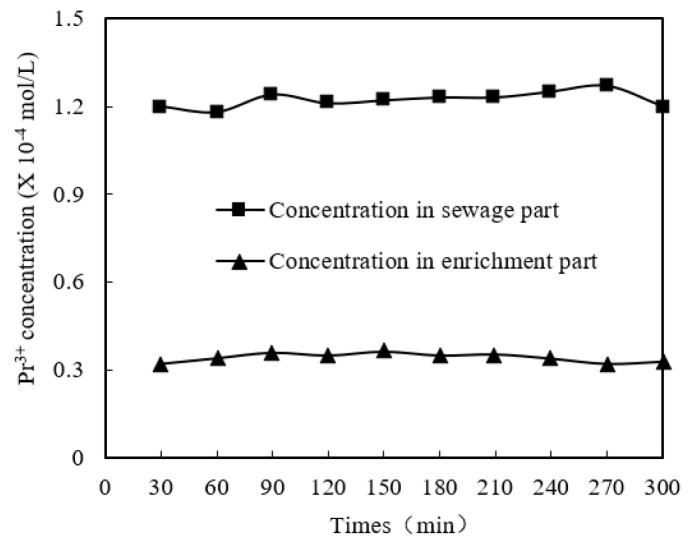
The constancy of MUFR.

**Figure 4 ijerph-19-09364-f004:**
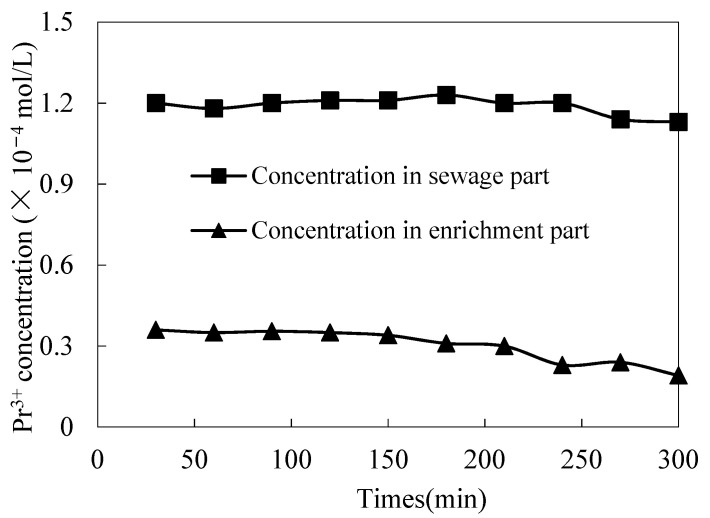
The constancy of conventional SLM.

**Figure 5 ijerph-19-09364-f005:**
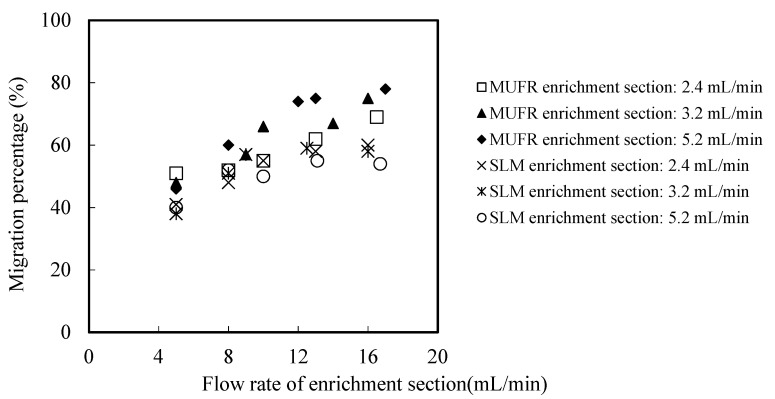
The comparison between MUFR and conventional SLM as regards flow velocity’s effect on migration of praseodymium(III) in both sections (I).

**Figure 6 ijerph-19-09364-f006:**
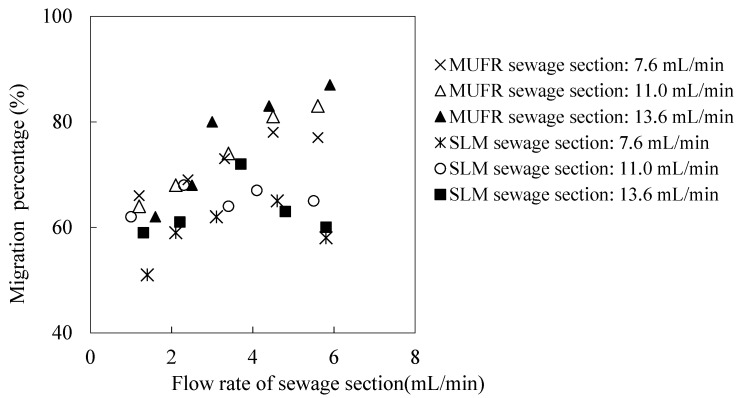
The comparison between MUFR and conventional SLM as regards flow velocity’s effect on the migration of praseodymium(III) in both sections (II).

**Figure 7 ijerph-19-09364-f007:**
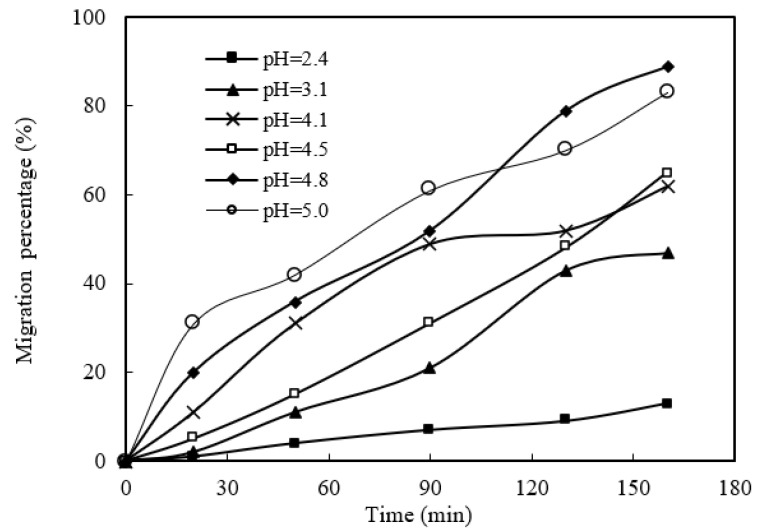
Effect of sewage pH.

**Figure 8 ijerph-19-09364-f008:**
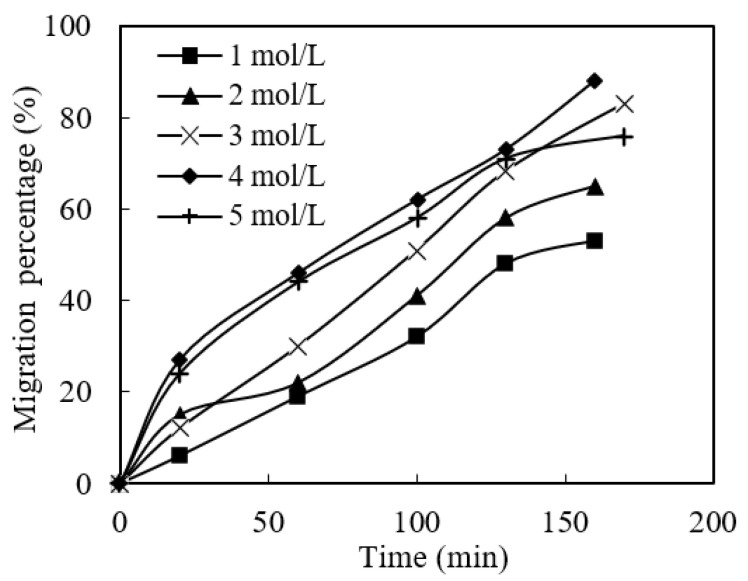
Effect of aqua fortis molarity on migration of Pr(III).

**Figure 9 ijerph-19-09364-f009:**
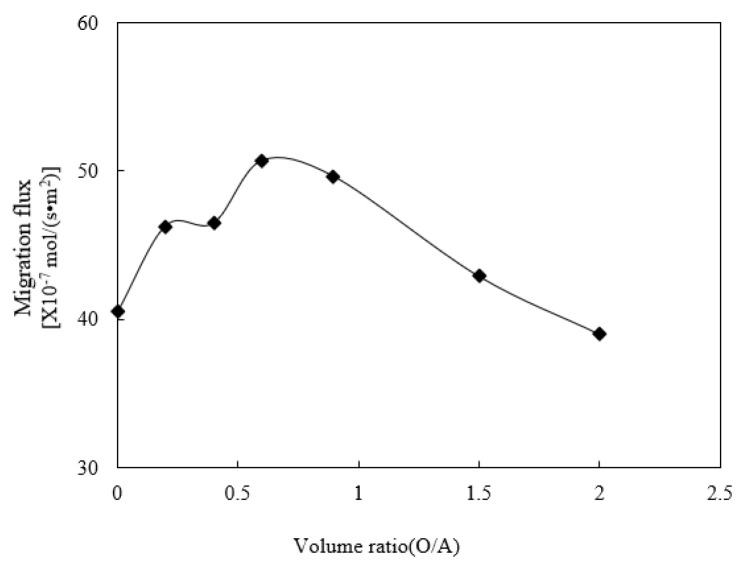
Effect of volume ratio of organophosphonic liquid to acid liquid (O/A).

**Figure 10 ijerph-19-09364-f010:**
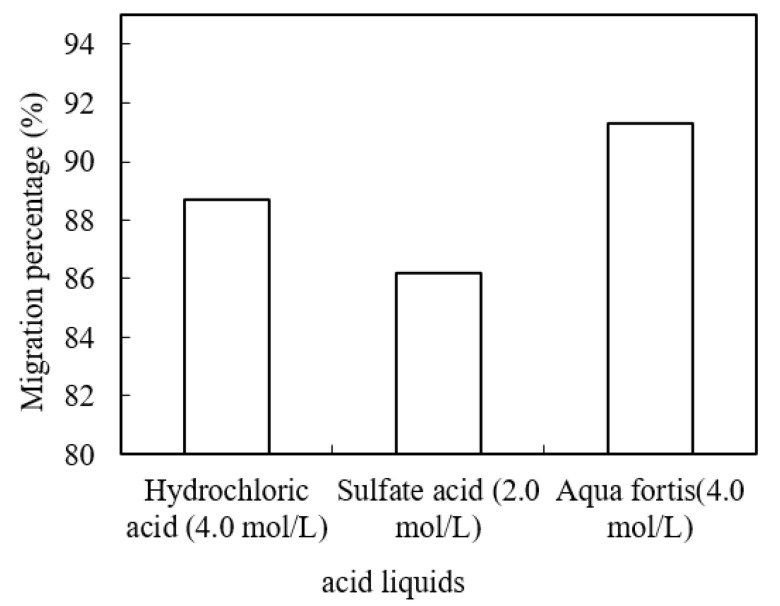
Effects of using different acid liquids as stripping agents.

**Figure 11 ijerph-19-09364-f011:**
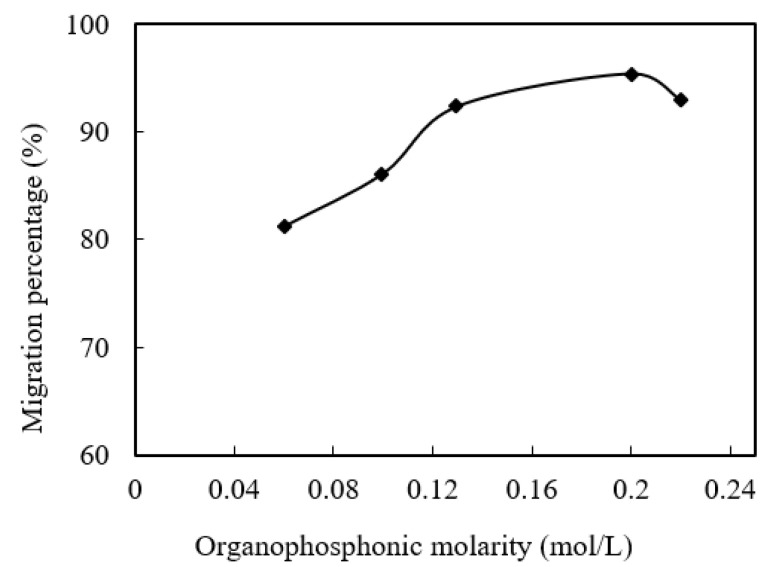
Effect of organophosphonic liquid on molarity.

**Figure 12 ijerph-19-09364-f012:**
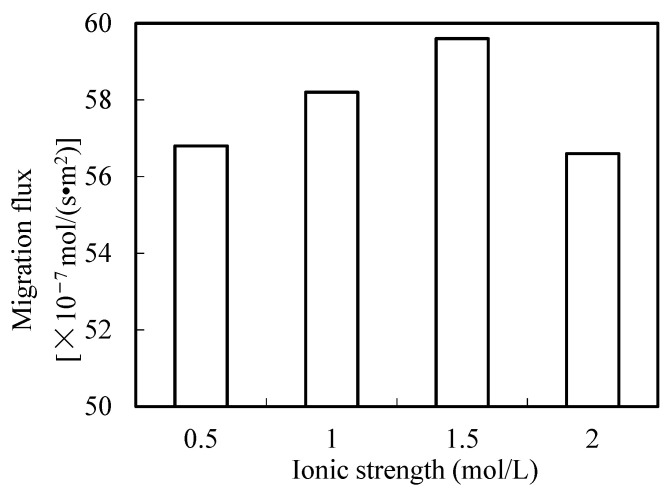
Effect of ionic strengths on the migration of praseodymium (III).

**Figure 13 ijerph-19-09364-f013:**
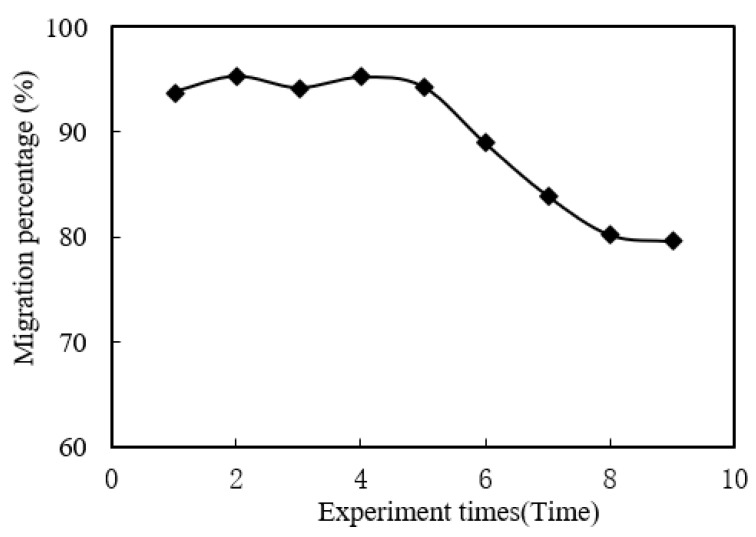
Reuse of organophosphonic liquid.

**Figure 14 ijerph-19-09364-f014:**
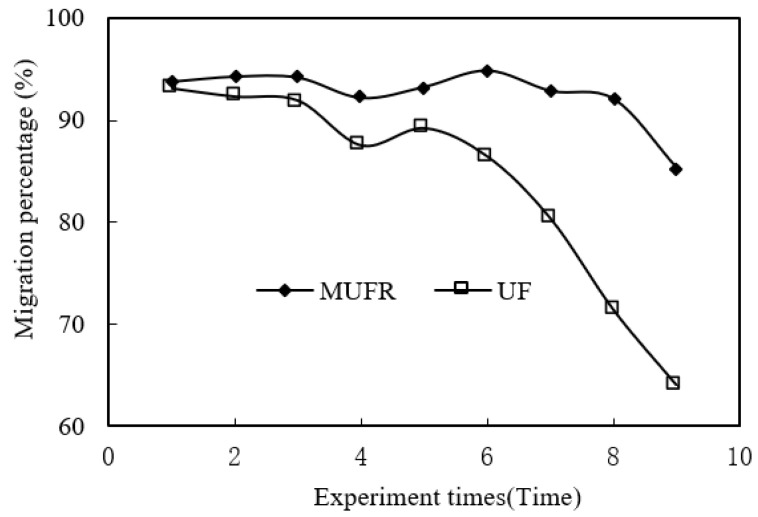
Reuse effect of PAN microtubules.

**Figure 15 ijerph-19-09364-f015:**
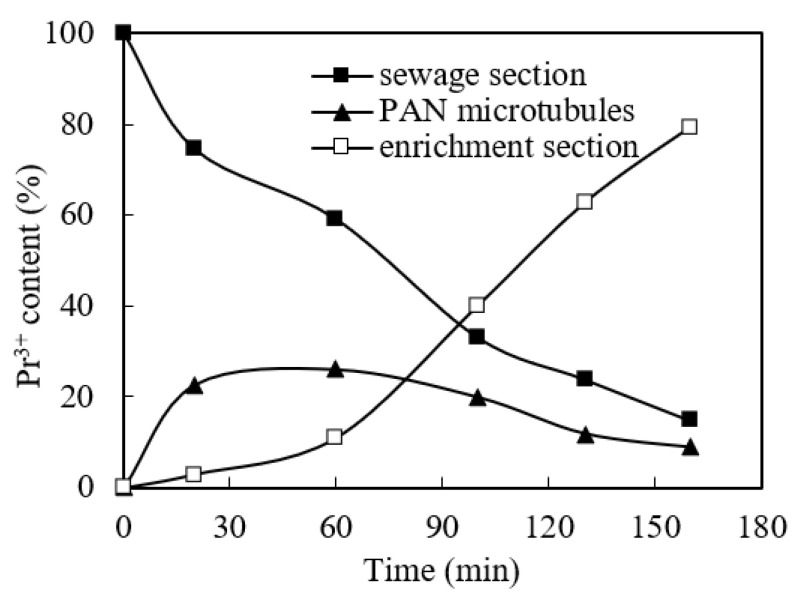
Retention in PAN microtubules and stripping effects.

**Figure 16 ijerph-19-09364-f016:**
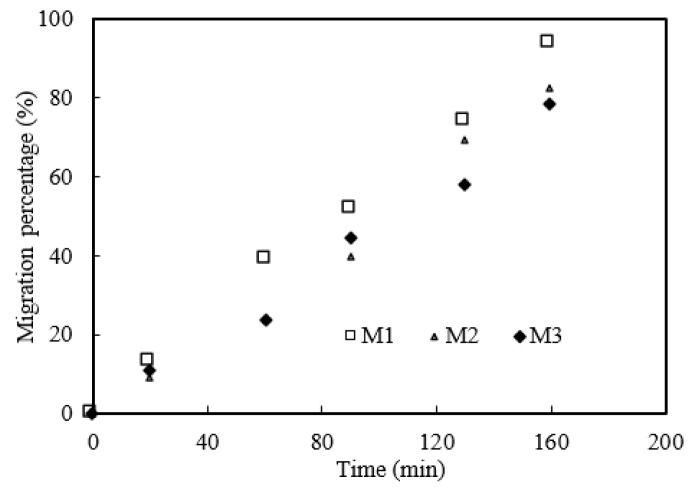
Effect of microtubule thickness.

**Figure 17 ijerph-19-09364-f017:**
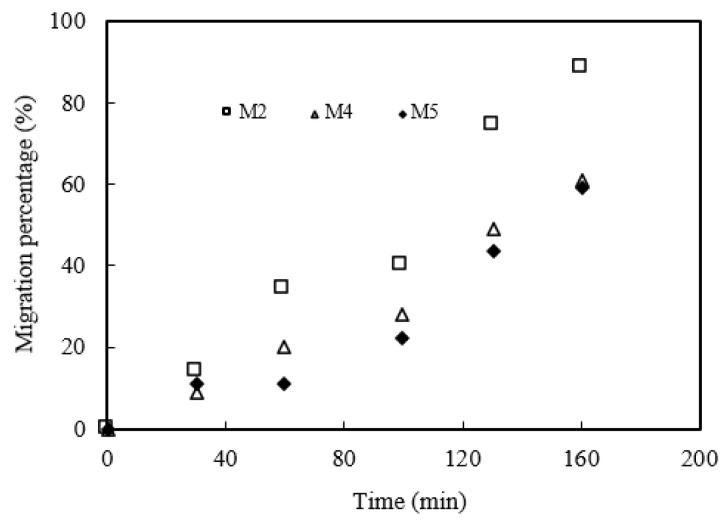
Effect of microtubule internal diameter.

**Figure 18 ijerph-19-09364-f018:**
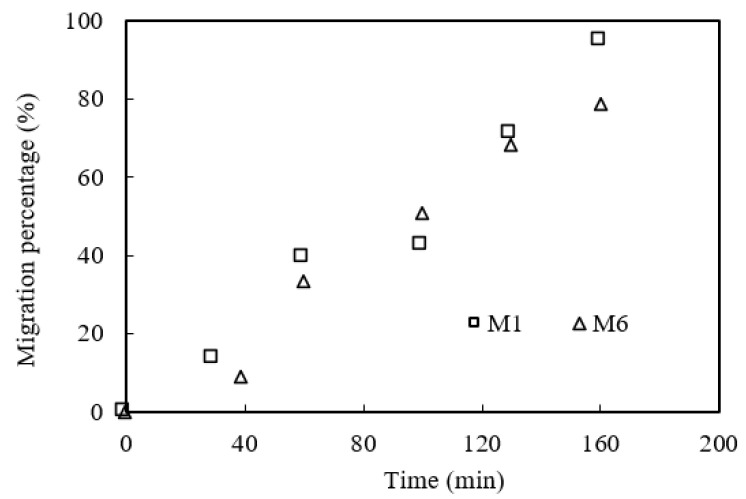
Effect of microtubule porosity.

**Table 1 ijerph-19-09364-t001:** Effect of praseodymium(III) cinit.

Time (min)	Migration Percentage (%)				
	8.00 × 10^−5^ mol/L	1.02 × 10^−4^ mol/L	1.29 × 10^−4^ mol/L	1.58 × 10^−4^ mol/L	2.00 × 10^−4^ mol/L
0	0	0	0	0	0
20	41.50	39.70	24.80	27.50	17.30
60	79.80	54.60	52.10	36.70	19.90
100	91.40	73.20	72.30	61.20	49.30
130	-	87.30	82.40	75.30	59.50
160	-	-	95.20	89.20	69.00

**Table 2 ijerph-19-09364-t002:** Structure parameters of microtubule module.

No.	PAN Microtubule Structure Parameters				
	Effective Length of Microtubule, *L*/(m)	Porosity	Number of Microtubules	Internal Diameter of Microtubule, *d*_i_/(mm)	Thickness of Microtubule, *d*_m_/(mm)
M1	0.25	63%	20	2.88	0.27
M2	0.25	63%	22	2.88	0.51
M3	0.25	63%	20	2.88	0.58
M4	0.25	63%	22	2.08	0.50
M5	0.25	63%	20	1.43	0.40
M6	0.25	22%	23	2.88	0.28

## Data Availability

Not applicable.

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
