# Peer review of "Migration of Trivalent Praseodymium from Tombarthite Sewage by Microtubule Ultrafiltration Reactor with Organophosphorus in Fuel Oil"

_ijerph, 2022, doi:10.3390/ijerph19159364_

Round 1

Reviewer 1 Report

The authors have done good study and therefore the paper may be considered for publication. However, the authors may like to address the following points in revision.

1.     Introduction needs improvement. It should have proper linkage with objectives of the study.

2.     Whether role of temperature and pressure has also been examined? If so, same may please be mentioned.

3.     How curvature of microtubes influences migration flux? This aspect should also be elaborated.

4.     If possible, quantify rate of flow velocity. Simply stating low or high velocity appears vague.

5.     What is meant by ‘Stronger power’ (line 328)?

6.     Please recast line 336-337.

7.     Applied aspect of results of this study should be highlighted to enhance value of authors contribution.

Author Response

Response to Reviewer 1:

Comments and Suggestions for Authors

The authors have done good study and therefore the paper may be considered for publication. However, the authors may like to address the following points in revision.

  1. Introduction needs improvement. It should have proper linkage with objectives of the study.

Response: Thank you for your advice.  Introduction has been improved, please see the places modified in the article.

  1. Whether role of temperature and pressure has also been examined? If so, same may please be mentioned.

Response: Thank you for your advice. Temperature and pressure has not been mentioned. Temperature and pressure have little effect on the system. This research does not need pressure as a driving force.

  1. How curvature of microtubes influences migration flux? This aspect should also be elaborated.

Response: Thank you for your advice.The curvature of microtubes influences migration flux, However, it is difficult to consider this factor in this study. There are many microtubules in ultrafiltration, and the curvature of each tube is slightly different. At present, we cannot homogenize to determine the curvature. In the future, we will consider the curvature factor.

  1. If possible, quantify rate of flow velocity. Simply stating low or high velocity appears vague.

Response: Thank you for your advice. We quantified the flow rate in Section 3.2. Please see the red words in 3.2.

  1. What is meant by ‘Stronger power’ (line 328)?

Response: Stronger power means “Strong hydrogen ion concentration difference”, because this concentration difference is the main driving force of material transmission. We added relevant descriptions in the text.

  1. Please recast line 336-337.

Response: Thank you for your advice. We have made changes.

  1. Applied aspect of results of this study should be highlighted to enhance value of authors contribution.

Response: Thank you for your advice. We put forward the application prospect at the conclusion of the article.

Reviewer 2 Report

Microtubule ultrafiltration reactor(MUFR) with organophosphorus system containing sewage section with buffer liquid acetic acid and enrichment section with aqua fortis liquid andorganophosphorus dissolved in fuel oil, was studied for the Praseodymium(III) migration in this paper. I think it is interesting for Hydrometallurgy researchers.

The paper might be publishable in light of a reasonable response/implementation of the following remarks:

1. How is it different from other processes, such as exchange, solvent extraction??

2. what about the efficiency in practice of MUFR.

3. The reaction in Figure 2, How is this information validated?

4. how much pressure in tube?

Author Response

Response to Reviewer 2:

Comments and Suggestions for Authors

Microtubule ultrafiltration reactor(MUFR) with organophosphorus system containing sewage section with buffer liquid acetic acid and enrichment section with aqua fortis liquid andorganophosphorus dissolved in fuel oil, was studied for the Praseodymium(III) migration in this paper. I think it is interesting for Hydrometallurgy researchers.

The paper might be publishable in light of a reasonable response/implementation of the following remarks:

  1. How is it different from other processes, such as exchange, solvent extraction??

Response: Thank you for your question. The process of this study is based on Extraction and supported liquid membrane. Supported liquid membrane is one more layer than extraction. In this study, we also used supported liquid membrane process, but we innovatively used it to ultrafilter microtubule devices. Few people have studied this in such detail. The results of this study show that the efficiency of our method is higher than that of traditional supported liquid membrane and much higher than that of extraction.

  1. what about the efficiency in practice of MUFR.

Response: Thank you for your question. This method is still in the laboratory stage, and there is still some distance from practical application. We add some expectations for future applications to the conclusion. Please see the red words in the conclusion part..

  1. The reaction in Figure 2, How is this information validated?

Response: Thank you for your question.Similar principles are introduced in references 11-14. This is also the basic principle of extracting and supporting liquid membrane.

  1. how much pressure in tube?

Response: Thank you for your question. Pressure has not been mentioned. Pressure have little effect on the system. This research does not need pressure as a driving force. The driving forces of system are hydrogen ion concentration difference, etc . So we didn't measure the pressure.

Reviewer 3 Report

I think that this article has a very good, modern topic in metallurgical area,  especially recovery of valuable metals (critical materials as  praseodymium(III) - rare earth metal). The article is the article contains the principle of sustainable development. The article a new method of migration of  praseodymium(III) ion was presented. In that of article little technology mistakes and typing errors were had. Anyway:

- device parameters should be added, e.g. volumes - section 2;

- Praseodymium (III) should be improved for praseodymium(III) in throughout the article (e.g. page 5 lines 173, 160 and 159)

- explain abbreviations in throughout the article (e.g.  Arsenazo III, MUFR)

-improve superscripts and subscripts (e.g. page 6 line 212, page 7 line 234)

- horizontal axis figures 3, 4, 13 and 14 should be supplemented by the unit e.g. times, h

- horizontal axis figure 9 should be supplemented by the unit e.g. O/A

-the significant places in table 2 shall be completed e.g. with 0.5 on 0.50 and with 0.4 for 0.40

- the same time unit in  throughout the article  e.g. hour or minute - author's choice

-the figure 8, 6, 5  must be improved  - are poorly readable

I recommended this article after minor correction to print.

Author Response

Response to Reviewer 3:

Comments and Suggestions for Authors

I think that this article has a very good, modern topic in metallurgical area,  especially recovery of valuable metals (critical materials as  praseodymium(III) - rare earth metal). The article is the article contains the principle of sustainable development. The article a new method of migration of  praseodymium(III) ion was presented. In that of article little technology mistakes and typing errors were had. Anyway:

1- device parameters should be added, e.g. volumes - section 2;

Response: Thank you for your advice. We added these. Please see the red words.

2- Praseodymium (III) should be improved for praseodymium(III) in throughout the article (e.g. page 5 lines 173, 160 and 159)

Response: Thank you for your advice. We have made all the modifications.Please see the red words.

3- explain abbreviations in throughout the article (e.g.  Arsenazo III, MUFR)

Response: Thank you for your advice. Arsenazo III is the name of common chemical reagent, not abbreviation. MUFR is microtubule ultrafiltration reactor. There are basically explanations in the article. If the editorial department needs, we can add an explanation table about symbol abbreviation at the end of the article.

4-improve superscripts and subscripts (e.g. page 6 line 212, page 7 line 234)

Response: Thank you for your advice. We have made all the modifications.Please see the red words.

5- horizontal axis figures 3, 4, 13 and 14 should be supplemented by the unit e.g. times, h

Response: Thank you for your advice. We have made all the modifications of these Figuers.

6- horizontal axis figure 9 should be supplemented by the unit e.g. O/A

Response: Thank you for your advice. We have made the modification of this figuer.

7-the significant places in table 2 shall be completed e.g. with 0.5 on 0.50 and with 0.4 for 0.40

Response: Thank you for your advice. We have made the modification of this table2 and table 1.

8-the same time unit in  throughout the article  e.g. hour or minute - author's choice

Response: Thank you for your advice. We have made the modification. Please see the figure 3,4.

9-the figure 8, 6, 5  must be improved  - are poorly readable. I recommended this article after minor correction to print.

 Response: Thank you for your advice. We have made the modification,Please see the figure 8,5,6.

Round 2

Reviewer 1 Report

By and large points have been taken care in revised version.